# Super-Resolution Off the Grid

**Qingqing Huang**
MIT,
EECS,
LIDS,
qqh@mit.edu

**Sham M. Kakade**
University of Washington,
Department of Statistics,
Computer Science & Engineering,
sham@cs.washington.edu

## Abstract

Super-resolution is the problem of recovering a superposition of point sources using bandlimited measurements, which may be corrupted with noise. This signal processing problem arises in numerous imaging problems, ranging from astronomy to biology to spectroscopy, where it is common to take (coarse) Fourier measurements of an object. Of particular interest is in obtaining estimation procedures which are robust to noise, with the following desirable statistical and computational properties: we seek to use coarse Fourier measurements (bounded by some *cutoff frequency*); we hope to take a (quantifiably) small number of measurements; we desire our algorithm to run quickly.

Suppose we have $k$ point sources in $d$ dimensions, where the points are separated by at least $\Delta$ from each other (in Euclidean distance). This work provides an algorithm with the following favorable guarantees:

- The algorithm uses Fourier measurements, whose frequencies are bounded by $O(1/\Delta)$ (up to log factors). Previous algorithms require a *cutoff frequency* which may be as large as $\Omega(\sqrt{d}/\Delta)$.
- The number of measurements taken by and the computational complexity of our algorithm are bounded by a polynomial in both the number of points $k$ and the dimension $d$, with *no* dependence on the separation $\Delta$. In contrast, previous algorithms depended inverse polynomially on the minimal separation and exponentially on the dimension for both of these quantities.

Our estimation procedure itself is simple: we take random bandlimited measurements (as opposed to taking an exponential number of measurements on the hypergrid). Furthermore, our analysis and algorithm are elementary (based on concentration bounds for sampling and the singular value decomposition).

## 1  Introduction

We follow the standard mathematical abstraction of this problem (Candes & Fernandez-Granda [4, 3]): consider a $d$-dimensional signal $x(t)$ modeled as a weighted sum of $k$ Dirac measures in $\mathbb{R}^d$:

$$x(t) = \sum_{j=1}^{k} w_j \delta_{\mu^{(j)}}, \tag{1}$$

where the point sources, the $\mu^{(j)}$'s, are in $\mathbb{R}^d$. Assume that the weights $w_j$ are complex valued, whose absolute values are lower and upper bounded by some positive constant. Assume that we are given $k$, the number of point sources[1].

Define the measurement function $f(s) : \mathbb{R}^d \to \mathbb{C}$ to be the convolution of the point source $x(t)$ with a low-pass point spread function $e^{i\pi<s,t>}$ as below:

$$f(s) = \int_{t\in\mathbb{R}^d} e^{i\pi<t,s>} x(\mathrm{d}t) = \sum_{j=1}^{k} w_j e^{i\pi<\mu^{(j)},s>}. \tag{2}$$

In the noisy setting, the measurements are corrupted by uniformly bounded perturbation $z$:

$$\widetilde{f}(s) = f(s) + z(s), \quad |z(s)| \le \epsilon_z, \forall s. \tag{3}$$

Suppose that we are only allowed to measure the signal $x(t)$ by evaluating the measurement function $\widetilde{f}(s)$ at any $s \in \mathbb{R}^d$, and we want to recover the parameters of the point source signal, i.e., $\{w_j, \mu^{(j)} : j \in [k]\}$. We follow the standard normalization to assume that:

$$\mu^{(j)} \in [-1, +1]^d, \quad |w_j| \in [0, 1] \quad \forall j \in [k].$$

Let $w_{min} = \min_j |w_j|$ denote the minimal weight, and let $\Delta$ be the minimal separation of the point sources defined as follows:

$$\Delta = \min_{j \neq j'} \|\mu^{(j)} - \mu^{(j')}\|_2, \tag{4}$$

where we use the Euclidean distance between the point sources for ease of exposition[2]. These quantities are key parameters in our algorithm and analysis. Intuitively, the recovery problem is harder if the minimal separation $\Delta$ is small and the minimal weight $w_{min}$ is small.

The first question is that, given exact measurements, namely $\epsilon_z = 0$, where and how many measurements should we take so that the original signal $x(t)$ can be exactly recovered.

**Definition 1.1** (Exact recovery). *In the exact case, i.e. $\epsilon_z = 0$, we say that an algorithm achieves exact recovery with $m$ measurements of the signal $x(t)$ if, upon input of these $m$ measurements, the algorithm returns the exact set of parameters $\{w_j, \mu^{(j)} : j \in [k]\}$.*

Moreover, we want the algorithm to be measurement noise tolerant, in the sense that in the presence of measurement noise we can still recover good estimates of the point sources.

**Definition 1.2** (Stable recovery). *In the noisy case, i.e., $\epsilon_z \ge 0$, we say that an algorithm achieves stable recovery with $m$ measurements of the signal $x(t)$ if, upon input of these $m$ measurements, the algorithm returns estimates $\{\widehat{w}_j, \widehat{\mu}^{(j)} : j \in [k]\}$ such that*

$$\min_{\pi} \max \left\{ \|\widehat{\mu}^{(j)} - \mu^{(\pi(j))}\|_2 : j \in [k] \right\} \le poly(d, k)\epsilon_z,$$

*where the $\min$ is over permutations $\pi$ on $[k]$ and poly(d,k) is a polynomial function in $d$ and $k$.*

By definition, if an algorithm achieves stable recovery with $m$ measurements, it also achieves exact recovery with these $m$ measurements.

The terminology of "super-resolution" is appropriate due to the following remarkable result (in the noiseless case) of Donoho [9]: suppose we want to accurately recover the point sources to an error of $\gamma$, where $\gamma \ll \Delta$. Naively, we may expect to require measurements whose frequency depends inversely on the desired the accuracy $\gamma$. Donoho [9] showed that it suffices to obtain a finite number of measurements, whose frequencies are bounded by $O(1/\Delta)$, in order to achieve *exact* recovery; thus resolving the point sources far more accurately than that which is naively implied by using frequencies of $O(1/\Delta)$. Furthermore, the work of Candes & Fernandez-Granda [4, 3] showed that stable recovery, in the univariate case ($d = 1$), is achievable with a cutoff frequency of $O(1/\Delta)$ using a convex program and a number of measurements whose size is polynomial in the relevant quantities.

| | d = 1 | | | d ≥ 1 | | |
|---|---|---|---|---|---|---|
| | cutoff freq | measurements | runtime | cutoff freq | measurements | runtime |
| SDP | $\frac{1}{\Delta}$ | $k\log(k)\log(\frac{1}{\Delta})$ | $poly(\frac{1}{\Delta},k)$ | $\frac{C_d}{\Delta_\infty}$ | $(\frac{1}{\Delta_\infty})^d$ | $poly((\frac{1}{\Delta_\infty})^d,k)$ |
| MP | $\frac{1}{\Delta}$ | $\frac{1}{\Delta}$ | $(\frac{1}{\Delta})^3$ | - | - | - |
| **Ours** | $\frac{1}{\Delta}$ | $(k\log(k))^2$ | $(k\log(k))^2$ | $\frac{\log(kd)}{\Delta}$ | $(k\log(k)+d)^2$ | $(k\log(k)+d)^2$ |

Table 1: See Section 1.2 for description. See Lemma 2.3 for details about the cutoff frequency. Here, we are implicitly using $O(\cdot)$ notation.

## 1.1 This work

We are interested in stable recovery procedures with the following desirable statistical and computational properties: we seek to use coarse (low frequency) measurements; we hope to take a (quantifiably) small number of measurements; we desire our algorithm run quickly. Informally, our main result is as follows:

**Theorem 1.3** (Informal statement of Theorem 2.2). *For a fixed probability of error, the proposed algorithm achieves stable recovery with a number of measurements and with computational runtime that are both on the order of $O((k\log(k)+d)^2)$. Furthermore, the algorithm makes measurements which are bounded in frequency by $O(1/\Delta)$ (ignoring log factors).*

Notably, our algorithm and analysis directly deal with the multivariate case, with the univariate case as a special case. Importantly, the number of measurements and the computational runtime do *not* depend on the minimal separation of the point sources. This may be important even in certain low dimensional imaging applications where taking physical measurements are costly (indeed, super-resolution is important in settings where $\Delta$ is small). Furthermore, our technical contribution of how to decompose a certain tensor constructed with Fourier measurements may be of broader interest to related questions in statistics, signal processing, and machine learning.

## 1.2 Comparison to related work

Table 1 summarizes the comparisons between our algorithm and the existing results. The multi-dimensional cutoff frequency we refer to in the table is the maximal coordinate-wise entry of any measurement frequency $s$ (i.e. $\|s\|_\infty$). "SDP" refers to the semidefinite programming (SDP) based algorithms of Candes & Fernandez-Granda [3, 4]; in the univariate case, the number of measurements can be reduced by the method in Tang et. al. [23] (this is reflected in the table). "MP" refers to the matrix pencil type of methods, studied in [14] and [15] for the univariate case. Here, we are defining the infinity norm separation as $\Delta_\infty = \min_{j\neq j'}\|\mu^{(j)}-\mu^{(j')}\|_\infty$, which is understood as the wrap around distance on the unit circle. $C_d \geq 1$ is a problem dependent constant (discussed below).

Observe the following differences between our algorithm and prior work:

1) Our minimal separation is measured under the $\ell_2$-norm instead of the infinity norm, as in the SDP based algorithm. Note that $\Delta_\infty$ depends on the coordinate system; in the worst case, it can underestimate the separation by a $1/\sqrt{d}$ factor, namely $\Delta_\infty \sim \Delta/\sqrt{d}$.

2) The computation complexity and number of measurements are polynomial in dimension $d$ and the number of point sources $k$, and surprisingly do not depend on the minimal separation of the point sources! Intuitively, when the minimal separation between the point sources is small, the problem should be harder, this is only reflected in the sampling range and the cutoff frequency of the measurements in our algorithm.

3) Furthermore, one could project the multivariate signal to the coordinates and solve multiple univariate problems (such as in [19, 17], which provided only exact recovery results). Naive random projections would lead to a cutoff frequency of $O(\sqrt{d}/\Delta)$.

**SDP approaches:** The work in [3, 4, 10] formulates the recovery problem as a total-variation min-imization problem; they then show the dual problem can be formulated as an SDP. They focused on the analysis of $d = 1$ and only explicitly extend the proofs for $d = 2$. For $d \geq 1$, Ingham-type theorems (see [20, 12]) suggest that $C_d = O(\sqrt{d})$.

The number of measurements can be reduced by the method in [23] for the $d = 1$ case, which is noted in the table. Their method uses sampling "off the grid"; technically, their sampling scheme is actually sampling random points from the grid, though with far fewer measurements.

**Matrix pencil approaches:** The matrix pencil method, MUSIC and Prony's method are essentially the same underlying idea, executed in different ways. The original Prony's method directly attempts to find roots of a high degree polynomial, where the root stability has few guarantees. Other methods aim to robustify the algorithm.

Recently, for the univariate matrix pencil method, Liao & Fannjiang [14] and Moitra [15] provide a stability analysis of the MUSIC algorithm. Moitra [15] studied the optimal relationship between the cutoff frequency and $\Delta$, showing that if the cutoff frequency is less than $1/\Delta$, then stable recovery is not possible with matrix pencil method (with high probability).

## 1.3   Notation

Let $\mathbb{R}$, $\mathbb{C}$, and $\mathbb{Z}$ to denote real, complex, and natural numbers. For $d \in \mathbb{Z}$, $[d]$ denotes the set $[d] = \{1, \ldots, d\}$. For a set $\mathcal{S}$, $|\mathcal{S}|$ denotes its cardinality. We use $\oplus$ to denote the direct sum of sets, namely $\mathcal{S}_1 \oplus \mathcal{S}_2 = \{(a + b) : a \in \mathcal{S}_1, b \in \mathcal{S}_2\}$.

Let $e_n$ to denote the $n$-th standard basis vector in $\mathbb{R}^d$, for $n \in [d]$. Let $\mathcal{P}^d_{R,2} = \{x \in \mathbb{R}^d : \|x\|_2 = 1\}$ to denote the $d$-sphere of radius $R$ in the $d$-dimensional standard Euclidean space.

Denote the condition number of a matrix $X \in \mathbb{R}^{m \times n}$ as $\text{cond}_2(X) = \sigma_{max}(X)/\sigma_{min}(X)$, where $\sigma_{max}(X)$ and $\sigma_{min}(X)$ are the maximal and minimal singular values of $X$.

We use $\otimes$ to denote tensor product. Given matrices $A, B, C \in \mathbb{C}^{m \times k}$, the tensor product $V = A \otimes B \otimes C \in \mathbb{C}^{m \times m \times m}$ is equivalent to $V_{i_1, i_2, i_3} = \sum_{n=1}^{k} A_{i_1, n} B_{i_2, n} C_{i_3, n}$. Another view of tensor is that it defines a multi-linear mapping. For given dimension $m_A, m_B, m_C$ the mapping $V(\cdot, \cdot, \cdot) : \mathbb{C}^{m \times m_A} \times \mathbb{C}^{m \times m_B} \times \mathbb{C}^{m \times m_C} \to \mathbb{C}^{m_A \times m_B \times m_C}$ is defined as:

$$[V(X_A, X_B, X_c)]_{i_1, i_2, i_3} = \sum_{j_1, j_2, j_3 \in [m]} V_{j_1, j_2, j_3}[X_A]_{j_1, i_1}[X_B]_{j_2, i_2}[X_C]_{j_3, i_3}.$$

In particular, for $a \in \mathbb{C}^m$, we use $V(I, I, a)$ to denote the projection of tensor $V$ along the 3rd dimension. Note that if the tensor admits a decomposition $V = A \otimes B \otimes C$, it is straightforward to verify that

$$V(I, I, a) = A Diag(C^\top a) B^\top.$$

It is well-known that if the factors $A, B, C$ have full column rank then the rank $k$ decomposition is unique up to re-scaling and common column permutation. Moreover, if the condition number of the factors are upper bounded by a positive constant, then one can compute the unique tensor decomposition $V$ with stability guarantees (See [1] for a review. Lemma 2.5 herein provides an explicit statement.).

## 2   Main Results

### 2.1   The algorithm

We briefly describe the steps of Algorithm 1 below:

**(Take measurements)** Given positive numbers $m$ and $R$, randomly draw a sampling set $\mathcal{S} = \{s^{(1)}, \ldots s^{(m)}\}$ of $m$ i.i.d. samples of the Gaussian distribution $\mathcal{N}(0, R^2 I_{d \times d})$. Form the set $\mathcal{S}' = \mathcal{S} \cup \{s^{(m+1)} = e_1, \ldots, s^{(m+d)} = e_d, s^{(m+d+1)} = 0\} \subset \mathbb{R}^d$. Denote $m' = m + d + 1$. Take another independent random sample $v$ from the unit sphere, and define $v^{(1)} = v$, $v^{(2)} = 2v$.

---

**Input:** $R$, $m$, noisy measurement function $\widetilde{f}(\cdot)$.

**Output:** Estimates $\{\widehat{w}_j, \widehat{\mu}^{(j)} : j \in [k]\}$.

1. **Take measurements:**

   Let $\mathcal{S} = \{s^{(1)}, \ldots, s^{(m)}\}$ be $m$ i.i.d. samples from the Gaussian distribution $\mathcal{N}(0, R^2 I_{d \times d})$.

   Set $s^{(m+n)} = e_n$ for all $n \in [d]$ and $s^{(m+n+1)} = 0$. Denote $m' = m + d + 1$.

   Take another random samples $v$ from the unit sphere, and set $v^{(1)} = v$ and $v^{(2)} = 2v$.

   Construct a tensor $\widetilde{F} \in \mathbb{C}^{m' \times m' \times 3}$: $\widetilde{F}_{n_1, n_2, n_3} = \widetilde{f}(s)\big|_{s = s^{(n_1)} + s^{(n_2)} + v^{(n_3)}}$.

2. **Tensor Decomposition:** Set $(\widehat{V}_{S'}, \widehat{D}_w) = \text{TensorDecomp}(\widetilde{F})$.

   For $j = 1, \ldots, k$, set $[\widehat{V}_{S'}]_j = [\widehat{V}_{S'}]_j / [\widehat{V}_{S'}]_{m', j}$

3. **Read of estimates:** For $j = 1, \ldots, k$, set $\widehat{\mu}^{(j)} = Real(\log([\widehat{V}_S]_{[m+1:m+d, j]}) / (i\pi))$.

4. Set $\widehat{W} = \arg\min_{W \in \mathbb{C}^k} \|\widehat{F} - \widehat{V}_{S'} \otimes \widehat{V}_{S'} \otimes \widehat{V}_d D_w\|_F$.

---

**Algorithm 1:** General algorithm

Construct the 3rd order tensor $\widetilde{F} \in \mathbb{C}^{m' \times m' \times 3}$ with noise corrupted measurements $\widetilde{f}(s)$ evaluated at the points in $\mathcal{S}' \oplus \mathcal{S}' \oplus \{v^{(1)}, v^{(2)}\}$, arranged in the following way:

$$\widetilde{F}_{n_1, n_2, n_3} = \widetilde{f}(s)\big|_{s = s^{(n_1)} + s^{(n_2)} + v^{(n_3)}}, \forall n_1, n_2 \in [m'], n_3 \in [2]. \tag{5}$$

**(Tensor decomposition)** Define the *characteristic matrix* $V_S$ to be:

$$V_S = \begin{bmatrix} e^{i\pi<\mu^{(1)}, s^{(1)}>} & \ldots & e^{i\pi<\mu^{(k)}, s^{(1)}>} \\ e^{i\pi<\mu^{(1)}, s^{(2)}>} & \ldots & e^{i\pi<\mu^{(k)}, s^{(2)}>} \\ \vdots & \ldots & \vdots \\ e^{i\pi<\mu^{(1)}, s^{(m)}>} & \ldots & e^{i\pi<\mu^{(k)}, s^{(m)}>} \end{bmatrix}. \tag{6}$$

and define matrix $V' \in \mathbb{C}^{m' \times k}$ to be

$$V_{S'} = \begin{bmatrix} V_S \\ V_d \\ 1, \ldots, 1 \end{bmatrix}, \tag{7}$$

where $V_d \in \mathbb{C}^{d \times k}$ is defined in (17). Define

$$V_2 = \begin{bmatrix} e^{i\pi<\mu^{(1)}, v^{(1)}>} & \ldots & e^{i\pi<\mu^{(k)}, v^{(1)}>} \\ e^{i\pi<\mu^{(1)}, v^{(2)}>} & \ldots & e^{i\pi<\mu^{(k)}, v^{(2)}>} \\ 1 & \ldots & 1 \end{bmatrix}.$$

Note that in the exact case ($\epsilon_z = 0$) the tensor $F$ constructed in (5) admits a rank-$k$ decomposition:

$$F = V_{S'} \otimes V_{S'} \otimes (V_2 D_w), \tag{8}$$

Assume that $V_{S'}$ has full column rank, then this tensor decomposition is unique up to column permutation and rescaling with very high probability over the randomness of the random unit vector $v$. Since each element of $V_{S'}$ has unit norm, and we know that the last row of $V_{S'}$ and the last row of $V_2$ are all ones, there exists a proper scaling so that we can uniquely recover $w_j$'s and columns of $V_{S'}$ up to common permutation.

In this paper, we adopt Jennrich's algorithm (see Algorithm 2) for tensor decomposition. Other algorithms, for example tensor power method ([1]) and recursive projection ([24]), which are possibly more stable than Jennrich's algorithm, can also be applied here.

**(Read off estimates)** Let $\log(V_d)$ denote the element-wise logarithm of $V_d$. The estimates of the point sources are given by:

$$\left[\mu^{(1)}, \mu^{(2)}, \ldots, \mu^{(k)}\right] = \frac{\log(V_d)}{i\pi}.$$

---
**Input:** Tensor $\widetilde{F} \in \mathbb{C}^{m \times m \times 3}$, rank $k$.

**output:** Factor $\widehat{V} \in \mathbb{C}^{m \times k}$.

1. Compute the truncated SVD of $\widetilde{F}(I, I, e_1) = \widehat{P}\widehat{\Lambda}\widehat{P}^\top$ with the $k$ leading singular values.

2. Set $\widehat{E} = \widetilde{F}(\widehat{P}, \widehat{P}, I)$. Set $\widehat{E}_1 = \widehat{E}(I, I, e_1)$ and $\widehat{E}_2 = \widehat{E}(I, I, e_2)$.

3. Let the columns of $\widehat{U}$ be the eigenvectors of $\widehat{E}_1 \widehat{E}_2^{-1}$ corresponding to the $k$ eigenvalues with the largest absolute value.

4. Set $\widehat{V} = \sqrt{m}\widehat{P}\widehat{U}$.
---

**Algorithm 2:** TensorDecomp

**Remark 2.1.** *In the toy example, the simple algorithm corresponds to using the sampling set $\mathcal{S}' = \{e_1, \ldots, e_d\}$. The conventional univariate matrix pencil method corresponds to using the sampling set $\mathcal{S}' = \{0, 1, \ldots, m\}$ and the set of measurements $\mathcal{S}' \oplus \mathcal{S}' \oplus \mathcal{S}'$ corresponds to the grid $[m]^3$.*

## 2.2 Guarantees

In this section, we discuss how to pick the two parameters $m$ and $R$ and prove that the proposed algorithm indeed achieves stable recovery in the presence of measurement noise.

**Theorem 2.2** (Stable recovery). *There exists a universal constant $C$ such that the following holds.*

*Fix $\epsilon_x, \delta_s, \delta_v \in (0, \frac{1}{2})$;*

*pick $m$ such that $m \geq \max\left\{ \frac{k}{\epsilon_x}\sqrt{8\log\frac{k}{\delta_s}}, \ d \right\}$;*

*for $d = 1$, pick $R \geq \frac{\sqrt{2\log(1+2/\epsilon_x)}}{\pi\Delta}$; for $d \geq 2$, pick $R \geq \frac{\sqrt{2\log(k/\epsilon_x)}}{\pi\Delta}$.*

*Assume the bounded measurement noise model as in (3) and that $\epsilon_z \leq \frac{\Delta \delta_v w_{min}^2}{100\sqrt{d}k^5}\left(\frac{1-2\epsilon_x}{1+2\epsilon_x}\right)^{2.5}$.*

*With probability at least $(1-\delta_s)$ over the random sampling of $\mathcal{S}$, and with probability at least $(1-\delta_v)$ over the random projections in Algorithm 2, the proposed Algorithm 1 returns an estimation of the point source signal $\widehat{x}(t) = \sum_{j=1}^{k} \widehat{w}_j \widehat{\delta}_{\mu^{(j)}}$ with accuracy:*

$$\min_\pi \max\left\{ \|\widehat{\mu}^{(j)} - \mu^{(\pi(j))}\|_2 : j \in [k] \right\} \leq C\frac{\sqrt{d}k^5}{\Delta \delta_v}\frac{w_{max}}{w_{min}^2}\left(\frac{1+2\epsilon_x}{1-2\epsilon_x}\right)^{2.5}\epsilon_z,$$

*where the $\min$ is over permutations $\pi$ on $[k]$. Moreover, the proposed algorithm has time complexity in the order of $O((m')^3)$.*

The next lemma shows that essentially, with overwhelming probability, all the frequencies taken concentrate within the hyper-cube with cutoff frequency $R'$ on each coordinate, where $R'$ is comparable to $R$,

**Lemma 2.3** (The cutoff frequency). *For $d > 1$, with high probability, all of the $2(m')^2$ sampling frequencies in $\mathcal{S}' \oplus \mathcal{S}' \oplus \{v^{(1)}, v^{(2)}\}$ satisfy that $\|s^{(j_1)} + s^{(j_2)} + v^{(j_3)}\|_\infty \leq R'$, $\forall j_1, j_2 \in [m], j_3 \in [2]$, where the per-coordinate cutoff frequency is given by $R' = O(R\sqrt{\log md})$.*

*For $d = 1$ case, the cutoff frequency $R'$ can be made to be in the order of $R' = O(1/\Delta)$.*

**Remark 2.4** (Failure probability). *Overall, the failure probability consists of two pieces: $\delta_v$ for random projection of $v$, and $\delta_s$ for random sampling to ensure the bounded condition number of $V_S$. This may be boosed to arbitrarily high probability through repetition.*

## 2.3 Key Lemmas

**Stability of tensor decomposition:** In this paragraph, we give a brief description and the stability guarantee of the well-known Jennrich's algorithm ([11, 13]) for low rank 3rd order tensor decomposition. We only state it for the symmetric tensors as appeared in the proposed algorithm.

Consider a tensor $F = V \otimes V \otimes (V_2 D_w) \in \mathbb{C}^{m \times m \times 3}$ where the factor $V$ has full column rank $k$. Then the decomposition is unique up to column permutation and rescaling, and Algorithm 2 finds the factors efficiently. Moreover, the eigen-decomposition is stable if the factor $V$ is well-conditioned and the eigenvalues of $F_a F_b^\dagger$ are well separated.

**Lemma 2.5** (Stability of Jennrich's algorithm). *Consider the 3rd order tensor $F = V \otimes V \otimes (V_2 D_w) \in \mathbb{C}^{m \times m \times 3}$ of rank $k \leq m$, constructed as in Step 1 in Algorithm 1.*

*Given a tensor $\widetilde{F}$ that is element-wise close to $F$, namely for all $n_1, n_2, n_3 \in [m]$, $\left| \widetilde{F}_{n_1, n_2, n_3} - F_{n_1, n_2, n_3} \right| \leq \epsilon_z$, and assume that the noise is small $\epsilon_z \leq \frac{\Delta \delta_v w_{min}^2}{100 \sqrt{d k} w_{max} cond_2(V)^5}$. Use $\widetilde{F}$ as the input to Algorithm 2. With probability at least $(1 - \delta_v)$ over the random projections $v^{(1)}$ and $v^{(2)}$, we can bound the distance between columns of the output $\widehat{V}$ and that of $V$ by:*

$$\min_{\pi} \max_{j} \left\{ \| \widehat{V}_j - V_{\pi(j)} \|_2 : j \in [k] \right\} \leq C \frac{\sqrt{d} k^2}{\Delta \delta_v} \frac{w_{max}}{w_{min}^2} cond_2(V)^5 \epsilon_z, \tag{9}$$

*where $C$ is a universal constant.*

**Condition number of $V_{S'}$:** The following lemma is helpful:

**Lemma 2.6.** *Let $V_{S'} \in \mathbb{C}^{(m+d+1) \times k}$ be the factor as defined in (7). Recall that $V_{S'} = [V_S; V_d; 1]$, where $V_d$ is defined in (17), and $V_S$ is the characteristic matrix defined in (6).*

*We can bound the condition number of $V_{S'}$ by*

$$cond_2(V_{S'}) \leq \sqrt{1 + \sqrt{k}} cond_2(V_S). \tag{10}$$

**Condition number of the characteristic matrix $V_S$:** Therefore, the stability analysis of the proposed algorithm boils down to understanding the relation between the random sampling set $\mathcal{S}$ and the condition number of the characteristic matrix $V_S$. This is analyzed in Lemma 2.8 (main technical lemma).

**Lemma 2.7.** *For any fixed number $\epsilon_x \in (0, 1/2)$. Consider a Gaussian vector $s$ with distribution $\mathcal{N}(0, R^2 I_{d \times d})$, where $R \geq \frac{\sqrt{2 \log(k/\epsilon_x)}}{\pi \Delta}$ for $d \geq 2$, and $R \geq \frac{\sqrt{2 \log(1 + 2/\epsilon_x)}}{\pi \Delta}$ for $d = 1$. Define the Hermitian random matrix $X_s \in \mathbb{C}_{herm}^{k \times k}$ to be*

$$X_s = \begin{bmatrix} e^{-i\pi <\mu^{(1)}, s>} \\ e^{-i\pi <\mu^{(2)}, s>} \\ \vdots \\ e^{-i\pi <\mu^{(k)}, s>} \end{bmatrix} \left[ e^{i\pi <\mu^{(1)}, s>}, e^{i\pi <\mu^{(2)}, s>}, \dots e^{i\pi <\mu^{(k)}, s>} \right]. \tag{11}$$

*We can bound the spectrum of $\mathbb{E}_s[X_s]$ by:*

$$(1 - \epsilon_x) I_{k \times k} \preceq \mathbb{E}_s[X_s] \preceq (1 + \epsilon_x) I_{k \times k}. \tag{12}$$

**Lemma 2.8** (Main technical lemma). *In the same setting of Lemma 2.7, Let $\mathcal{S} = \{s^{(1)}, \dots, s^{(m)}\}$ be $m$ independent samples of the Gaussian vector $s$. For $m \geq \frac{k}{\epsilon_x} \sqrt{8 \log \frac{k}{\delta_s}}$, with probability at least $1 - \delta_s$ over the random sampling, the condition number of the factor $V_S$ is bounded by:*

$$cond_2(V_S) \leq \sqrt{\frac{1 + 2\epsilon_x}{1 - 2\epsilon_x}}. \tag{13}$$

**Acknowledgments**

The authors thank Rong Ge and Ankur Moitra for very helpful discussions. Sham Kakade acknowledges funding from the Washington Research Foundation for innovation in Data-intensive Discovery.

## Footnotes

[1] An upper bound of the number of point sources suffices.

[2]Our claims hold withut using the "wrap around metric", as in [4, 3], due to our random sampling. Also, it is possible to extend these results for the $\ell_p$-norm case.

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
