[Supplementary Material · multi-prony-nips-supp.pdf]

# 3 Warm-up

## 3.1 1-D case: revisiting the matrix pencil method

Let us first review the matrix pencil method for the univariate case, which stability was recently rigorously analyzed in Liao & Fannjiang [14] and Moitra [15].

A square matrix $H$ is called a *Hankel* matrix if its skew-diagonals are constants, namely $H_{i,j} = H_{i-1,j+1}$. For some positive constants $m \in \mathbb{Z}$, sample to get the measurements $f(s)$ evaluated at the sampling set $\mathcal{S}_3 = \{0, 1, \ldots, 2m\}$, and construct two Hankel matrices $H_0, H_1 \in \mathbb{C}^{m \times m}$:

$$H_0 = \begin{bmatrix} f(0) & f(1) & \ldots & f(m-1) \\ f(1) & f(2) & \ldots & f(m) \\ \vdots & & & \vdots \\ f(m-1) & f(m) & \ldots & f(2m-1) \end{bmatrix}, \quad H_1 = \begin{bmatrix} f(1) & f(2) & \ldots & f(m) \\ f(2) & f(3) & \ldots & f(m+1) \\ \vdots & & & \vdots \\ f(m) & f(m+1) & \ldots & f(2m) \end{bmatrix}. \tag{14}$$

Define $D_w \in \mathbb{C}_{diag}^{k \times k}$ to be the diagonal matrix with the weights on the main diagonal: $[D_w]_{j,j} = w_j$. Define $D_\mu \in \mathbb{C}_{diag}^{k \times k}$ to be $[D_\mu]_{j,j} = e^{i\pi\mu^{(j)}}$.

A matrix $V$ is called a *Vandermonde matrix* if each column is a geometric progression. defined the Vandermonde matrix $V_m \in \mathbb{C}^{m \times k}$ as below:

$$V_m = \begin{bmatrix} 1 & \ldots & 1 \\ (e^{i\pi\mu^{(1)}})^1 & \ldots & (e^{i\pi\mu^{(k)}})^1 \\ \vdots & & \vdots \\ (e^{i\pi\mu^{(1)}})^{m-1} & \ldots & (e^{i\pi\mu^{(k)}})^{m-1} \end{bmatrix}. \tag{15}$$

The two Hankel matrices $H_0$ and $H_1$ admit the following simultaneous diagonalization:

$$H_0 = V_m D_w V_m^\top, \quad H_1 = V_m D_w D_\mu V_m^\top. \tag{16}$$

As long as $V_m$ is of full rank, this simultaneous diagonalization can be computed by solving the generalized eigenvalue problem, and the parameters of the point source can thus be obtained from the factor $V_m$ and $D_w$.

The univariate matrix pencil method only needs $m \geq k$ to achieve exact recovery. In the noisy case, the stability of generalized eigenvalue problem depends on the condition number of the Vandermonde matrix $V_m$ and the minimal weight $w_{min}$.

Since all the nodes ($e^{i\pi\mu^{(j)}}$'s) of this Vandermonde matrix lie on the unit circle in the complex plane, it is straightforward to see that asymptotically $\lim_{m \to \infty} \text{cond}_2(V_m) = 1$. Furthermore, for $m > 1/\Delta$, [14, 15] showed that $\text{cond}_2(V_m)$ is upper bounded by a constant that does not depend on $k$ and $m$. This bound on condition number is also implicitly discussed in [19].

Another way to view the matrix pencil method is that it corresponds to the low rank 3rd order tensor decomposition (see for example [1]). This view will help us generalize matrix pencil method to higher dimension $d$ in a direct way, without projecting the signal on each coordinate and apply the univariate algorithm multiple times. For $m \geq k$, construct a 3rd order tensor $F \in \mathbb{C}^{m \times m \times 2}$ with elements of $H_0$ and $H_1$ defined in (14) as:

$$F_{i,i',j} = [H_{j-1}]_{i,i'}, \quad \forall j \in [2], i, i' \in [m].$$

Note that the two slices along the 3rd dimension of $F$ are $H_0$ and $H_1$. Namely $F(I, I, e_1) = H_0$, and $F(I, I, e_2) = H_1$. Recall the matrix decomposition of $H_0$ and $H_1$ in (16). Since $m \geq k$ and the $\mu^{(j)}$'s are distinct, we know that $F$ has the *unique* rank $k$ tensor decomposition:

$$F = V_m \otimes V_m \otimes (V_2 D_w).$$

Given the tensor $F$, the basic idea of the well-known Jennrich's algorithm ([11, 13]) for finding the unique low rank tensor decomposition is to consider two random projections $v_1, v_2 \in \mathbb{R}^m$, and then with high probability the two matrices $F(I, I, v_1)$ and $F(I, I, v_2)$ admit simultaneous diagonalization. Therefore, the matrix pencil method is indeed a special case of Jennrich's algorithm by setting $v_1 = e_1$ and $v_2 = e_2$

## 3.2 The multivariate case: a toy example

One could naively extend the matrix pencil method to higher dimensions by taking measurements from a hyper-grid, which is of size exponential in the dimension $d$. We now examine a toy problem which suggests that the high dimensional case may not be inherently more difficult than the univariate case.

The key ideas is that an appropriately sampled set can significantly reduce the number of measurements (as compared to using all the grid points). Tang et al [23] made a similar observation for the univariate case. They used a small random subset of measurements (actually still from the grid points) and showed that this contains enough information to recover all the measurement on the grid; the full measurements were then used for stably recovering the point sources.

Consider the case where the dimension $d \geq k$. Assume that $w_j$'s are real valued, and for all $j \in [k]$ and $n \in [d]$, the parameters $\mu_n^{(j)}$ are i.i.d. and uniformly distributed over $[-1, +1]$. This essentially corresponds to the standard ($L_2$) incoherence conditions (for the $\mu^{(j)}$'s). [3] The following simple algorithm achieves stability with polynomial complexity.

First, take $d^3$ number of measurements by evaluating $f(s)$ in the set $\mathcal{S}_3 = \{ s = e_{n_1} + e_{n_2} + e_{n_3} : [n_1, n_2, n_3] \in [d] \times [d] \times [d] \}$, noting that $\mathcal{S}_3$ contains only a subset of $d^3$ points from the grid of $[3]^d$. Then, construct a 3rd order tensor $F \in \mathbb{C}^{d \times d \times d}$ with the measurements in the following way:

$$F_{n_1,n_2,n_3} = f(s)\big|_{s=e_{n_1}+e_{n_2}+e_{n_3}}, \quad \forall n_1, n_2, n_3 \in [d].$$

Note that the measurement $f(e_1 + e_2 + e_3) = \sum_{j=1}^{k} w_j e^{i\pi(\mu_1^{(j)}+\mu_2^{(j)}+\mu_3^{(j)})} = \sum_{j=1}^{k} w_j e^{i\pi\mu_1^{(j)}} e^{i\pi\mu_2^{(j)}} e^{i\pi\mu_3^{(j)}}$. It is straightforward to verify that $F$ has a rank-$k$ tensor factorization $F = V_d \otimes V_d \otimes (V_d D_w)$, where the factor $V_d \in \mathbb{R}^{d \times k}$ is given by:

$$V_d = \begin{bmatrix} e^{i\pi\mu_1^{(1)}} & \cdots & e^{i\pi\mu_1^{(k)}} \\ e^{i\pi\mu_2^{(1)}} & \cdots & e^{i\pi\mu_2^{(k)}} \\ \vdots & \cdots & \vdots \\ e^{i\pi\mu_d^{(1)}} & \cdots & e^{i\pi\mu_d^{(k)}} \end{bmatrix}. \tag{17}$$

Under the distribution assumption of the point sources, the entries $e^{i\pi\mu_n^{(j)}}$ are i.i.d. and uniformly distributed over the unit circle on the complex plane. Therefore almost surely the factor $V_d$ has full column rank, and thus the tensor decomposition is unique. Moreover here $w_j$'s are real and each element of $V_S$ has unit norm, we have a rescaling constraint with the tensor decomposition, with which we can uniquely obtain the factor $V_S$ and the weights in $D_w$. By taking element-wise log of $V_S$ we can read off the parameters of the point sources from $V_S$ directly. Moreover, with high probability, we have that $\mathrm{cond}_2(V_d)$ concentrates around 1, thus the simple algorithm achieves stable recovery.

## 4 Proofs

*Proof.* (of Theorem 2.2) The algorithm is correct if the tensor decomposition in Step 2 is unique, and achieves stable recovery if the tensor decomposition is stable. By the stability Lemma of tensor decomposition (Lemma 2.5), this is guaranteed if we can bound the condition number of $V_{S'}$. It follows from Lemma 2.6 that the condition number of $V_{S'}$ is at most $\sqrt{1 + \sqrt{k}}$ times of $\mathrm{cond}_2(V_S)$. By the main technical lemma (Lemma 2.8) we know that with the random sampling set $\mathcal{S}$ of size $m$, the condition number $\mathrm{cond}_2(V_S)$ is upper bounded by a constant. Thus we can bound the distance between $V_{S'}$ and the estimation $\widehat{V}_{S'}$ according to (9).

Since we adopt Jennrich's algorithm for the low rank tensor decomposition, the overall computation complexity is roughly the complexity of SVD of a matrix of size $m' \times m'$, namely in the order of $O((m')^3)$. □

*Proof.* (of Lemma 2.3) For $d > 1$ case, with straightforward union bound over the $m' = O(k^2)$ samples each of which has $d$ coordinates, one can show that the cutoff frequency is in the order of $R\sqrt{\log(kd)}$, where $R$ is in the order of $\frac{\sqrt{\log(k)}}{\Delta}$ as shown in Theorem 2.2.

For $d = 1$ case, we bound the cutoff frequency with slightly more careful analysis. Instead of Gaussian random samples, consider uniform samples from the interval $[-R', R']$. We can modify the proof of Lemma 2.7 and show that if $R' \geq 1/(\Delta(1 + \epsilon_x))$:

$$\sum_{j' \neq j} |Y_{j,j'}| = \sum_{j' \neq j} \frac{1}{2R'} \int_{-R',R'} e^{i\pi(\mu^{j'} - \mu^{(j)})s} = \sum_{j' \neq j} \frac{\sin(\pi|\mu^{(j')} - \mu^{(j)}|R')}{\pi|\mu^{(j')} - \mu^{(j)}|R'}$$

$$\leq \sum_{l=1}^{k} \frac{\sin(l\pi\Delta R')}{(l\pi\Delta R')} \leq \frac{\sin(\pi\Delta R')/(\pi\Delta R')}{1 - \sin(\pi\Delta R')/(\pi\Delta R')} \leq \epsilon_x$$

where the second last inequality uses the inequality that $\frac{\sin(a+b)}{a+b} \leq \frac{\sin(a)}{a}\frac{\sin(b)}{b}$. $\quad\square$

*Proof.* (of Lemma 2.5) The proof is mostly based on the arguments in [16, 2], we still show the clean arguments here for our case.

We first introduce some notations for the exact case. Define $D_1 = \text{diag}([V_2]_{1,:}D_w)$ and $D_2 = \text{diag}([V_2]_{2,:}D_w)$. Recall that the symmetric matrix $F_1 = F(I, I, e_1) = VD_1V^\top$. Consider its SVD $F_1 = P\Lambda P^\top$. Denote $U = P^\top V \in \mathbb{C}^{k \times k}$. Define the whitened rank-$k$ tensor

$$E = F(P, P, I) = (P^\top V) \otimes (P^\top V) \otimes (V_2 D_w) = U \otimes U \otimes (V_2 D_w) \in \mathbb{C}^{k \times k \times 3}.$$

Denote the two slices of the tensor $E$ by $E_1 = E(I, I, e_1) = UD_1U^\top$ and $E_2 = E(I, I, e_2) = UD_2U^\top$. Define $M = E_1E_2^{-1}$, and its eigen decomposition is given by $M = UDU^{-1}$, where $D = D_1D_2^{-1}$. Note that in the exact case, $D$ is given by:

$$D = \text{diag}(e^{i\pi < \mu^{(j)}, v^{(1)} - v^{(2)} >} : j \in [k])$$

Note that $|D_{j,j}| = 1$ for all $j$. Define the minimal separation of the diagonal entries in $D$ to be:

$$sep(D) = \min\{\min_{j \neq j'} |D_{j,j} - D_{j',j'}|\},$$

1. We first apply perturbation bounds to show that the noise in $\widetilde{F}$ propagates the estimates $\widehat{P}$ and $\widehat{E}$ in a mild way when the condition number of $V$ is bounded by a constant.

*Proof.* Apply Wedin's matrix perturbation bound, we have:

$$\|\widehat{P} - P\|_2 \leq \frac{\|\widetilde{F}_1 - F_1\|_2}{\sigma_{min}(F_1)} \leq \frac{\epsilon_z\sqrt{m}}{w_{min}\sigma_{min}(V)^2}$$

And then for the two slices of $\widehat{E} = \widetilde{F}(\widehat{P}, \widehat{P}, I)$, namely $\widehat{E}_i = E_i + Z_i$ for $i = 1, 2$, we can bound the distance between estimates and the exact case, namely $Z_i = \widehat{P}^\top \widetilde{F}_i \widehat{P} - P^\top F_i P$, by:

$$\|Z_i\| \leq 8\|F_i\|\|P\|\|\widehat{P} - P\| + 4\|P\|^2\|\widetilde{F}_i - F_i\| \leq 16\frac{w_{max}}{w_{min}}\text{cond}_2(V)^2\epsilon_z\sqrt{m}$$

$\quad\square$

2. Then, recall that $M = E_1E_2^{-1} = UDU^{-1}$. Note that

$$\widehat{M} = (E_1 + Z_1)(E_2 + Z_2)^{-1} = E_1E_2^{-1}(I - Z_2(I + E_2^{-1}Z_2)^{-1}E_2^{-1}) + Z_1E_2^{-1}.$$

Let $H$ and $G$ denote the perturbation matrices:

$$H = -Z_2(I + E_2^{-1}Z_2)^{-1}E_2^{-1}, \quad G = Z_1E_2^{-1}.$$

In the following claim, we show that given $\widehat{M} = \widehat{E}_1\widehat{E}_2^{-1} = M(I + H) + G$ for some small perturbation matrix $H$ and $G$, if the perturbation $\|H\|$ and $\|G\|$ are small enough and that $sep(D)$ is large enough, the eigen decomposition $\widehat{M} = \widehat{U}\widehat{D}\widehat{U}^{-1}$ is close to that of $M$.

**Claim 4.1.** *If* $\|MH + G\| \leq \frac{sep(D)}{2\sqrt{k}cond_2(U)}$, *then the eigenvalues of* $\widehat{M}$ *are distinct and we can bound the columns of* $\widehat{U}$ *and* $U$ *by:*

$$\min_{\pi} \max_j \|\widehat{U}_j - U_{\pi(j)}\|_2 \leq 3\frac{\sigma_{max}(H)\sigma_{max}(D) + \sigma_{max}(G)}{\sigma_{min}(U)sep(D)}\|\widehat{U}_j\|_2\|V_j\|_2.$$

*Proof.* Let $\lambda_j$ and $U_j$ for $j \in [k]$ denote the eigenvalue and corresponding eigenvectors of $M$. If $\|MH + G\| \leq \frac{sep(D)}{2\sqrt{k}cond_2(U)}$, we can bound

$$\|\widehat{M} - M\| = \|U^{-1}(M + (MH + G))U - D\| = \|U^{-1}(MH + G)U\| \leq sep(D)/2\sqrt{k},$$

thus apply Gershgorin's disk theorem, we have $|\widehat{\lambda}_j - \lambda_j| \leq \|[U^{-1}(MH + G)U]_j\|_1 \leq \sqrt{k}\|[U^{-1}(MH + G)U]_j\|_2 \leq sep(D)/2$. Therefore, the eigenvalues are distinct and we have

$$|\widehat{\lambda}_j - \lambda_{j'}| \geq |\lambda_j - \lambda_{j'}| - |\widehat{\lambda}_j - \lambda_j| \geq \frac{1}{2}|\lambda_j - \lambda_{j'}| \geq \frac{1}{2}sep(D). \tag{18}$$

Note that $\{U_{j'}\}$ and $\{\widehat{U}_j\}$ define two sets of basis vectors, thus we can write $\widehat{U}_j = \sum_{j'} c_{j'} U_{j'}$ (with the correct permutation for columns of $\widehat{U}_j$ and $U_j$) for some coefficients $\sum_{j'} c_{j'}^2 = 1$. Apply first order Taylor expansion of eigenvector definition we have:

$$\widehat{\lambda}_j\widehat{U}_j = \widehat{M}\widehat{U}_j = (M + (MH + G))\sum_{j'} c_{j'} U_{j'} = \sum_{j'} \lambda_{j'} c_{j'} U_{j'} + (MH + G)\widehat{U}_j.$$

Since we also have $\widehat{\lambda}_j\widehat{U}_j = \sum_{j'} \widehat{\lambda}_j c_{j'} U_{j'}$, we can write $\sum_{j'}(\widehat{\lambda}_j - \lambda_{j'})c_{j'} U_{j'} = (MH + G)\widehat{U}_j$, and we can solve for the coefficients $c_{j'}$'s from the linear system as $[(\widehat{\lambda}_j - \lambda_{j'})c_{j'} : j' \in [k]] = U^{-1}(MH + G)\widehat{U}_j$. Finally plug in the inequality in (18) we have that for any $j$:

$$\begin{aligned}
\|\widehat{U}_j - U_j\|_2^2 &= \sum_{j' \neq j} c_{j'}^2\|U_{j'}\|_2^2 + (c_j - 1)^2\|U_j\|_2^2 \\
&\leq 2\sum_{j' \neq j} c_{j'}^2\|V_{j'}\|_2^2 \\
&\leq 8\frac{\|U^{-1}(MH + G)\widehat{U}_j\|_2^2}{sep(D)^2} \\
&\leq 8\frac{(\sigma_{max}(D)\sigma_{max}(H) + \sigma_{max}(G))^2}{\sigma_{min}(U)^2 sep(D)^2}\|\widehat{U}_j\|_2^2\|V_j\|_2^2
\end{aligned}$$

$\square$

3. Note that in the above bound for $\|\widehat{U}_j - U_j\|$, we can bound the perturbation matrices $H$ and $G$ by:

$$\sigma_{max}(H) \leq \frac{\|Z_2\|}{(1 - \sigma_{max}(E_2^{-1}Z_2))\sigma_{min}(E_2)} \leq \frac{\|Z_2\|}{\sigma_{min}(E_2) - \|Z_2\|} \leq \frac{\|Z_2\|}{\sigma_{min}(U)^2\sigma_{min}(D_2) - \|Z_2\|},$$

$$\sigma_{max}(G) \leq \frac{\sigma_{max}(Z_1)}{\sigma_{min}(E_2)} \leq \frac{\|Z_2\|}{\sigma_{min}(U)^2\sigma_{min}(D_2)},$$

Note that $\sigma_{min}(D_2) \geq w_{min}$ and $\sigma_{max}(D) = 1$ by definition. In the following claim, we apply anti-concentration bound to show that with high probability $sep(D)$ is large.

**Claim 4.2.** *For any* $\delta_v \in (0, 1)$, *with probability at least* $1 - \delta_v$, *we can bound* $sep(D)$ *by:*

$$sep(D) \geq \frac{\Delta\delta_v}{\sqrt{d}k^2}.$$

*Proof.* Denote $v = v^{(1)} - v^{(2)}$, and note that $\|v\| \leq \sqrt{2}$. In the regime we concern, for any pair $j \neq j'$, we have $|e^{i\pi<\mu^{(j)},v>} - e^{i\pi<\mu^{(j')},v>}| \leq |<\mu^{(j)} - \mu^{(j')}, v>|$. Apply Lemma 5.3, we have that for $\delta \in (0, 1)$,

$$\mathbb{P}(|<\mu^{(j)} - \mu^{(j')}, v>| \leq \|\mu^{(j)} - \mu^{(j')}\| \frac{\delta}{\sqrt{d}}) \leq \delta.$$

Take a union bound over all pairs of $j \neq j'$, we have that

$$\mathbb{P}\left( \text{for some} j \neq j', |<\mu^{(j)} - \mu^{(j')}, v>| \leq \|\mu^{(j)} - \mu^{(j')}\| \frac{\delta}{\sqrt{d}k^2} \right) \leq k^2 \frac{\delta}{k^2} = \delta.$$

Recall that $\Delta = \min_{j \neq j'} \|\mu^{(j)} - \mu^{(j')}\|$. $\qquad\qquad\qquad\qquad\qquad\qquad\qquad\qquad\square$

4. Recall that $U = P^\top V$. Note that since $P$ has orthonormal columns, we have $\sigma_{min}(U) = \sigma_{min}(V)$ and $\|U_i\| \leq \|V_i\| = \sqrt{m}$.

Finally we apply perturbation bound to the estimates $\widehat{V}_i = \widehat{P}\widehat{U}_i$ and conclude with the above inequalities:

$$
\begin{aligned}
\|\widehat{V}_i - V_i\| &\leq 2(\|\widehat{P} - P\|\|U_i\| + \|P\|\|\widehat{U}_i - U_i\|) \\
&\leq 2\left( \frac{\epsilon_z \sqrt{m}}{w_{min}\sigma_{min}(V)^2} + 3\frac{\sigma_{max}(H)\sigma_{max}(D) + \sigma_{max}(G)}{\sigma_{min}(U)sep(D)}\|V_i\| \right)\|V_i\| \\
&\leq 2\left( \frac{\epsilon_z \sqrt{m}}{w_{min}\sigma_{min}(V)^2} + 6\frac{\|Z_2\|\|V_i\|}{(\sigma_{min}(V)^2\sigma_{min}(D_2) - \|Z_2\|)\sigma_{min}(V)sep(D)} \right)\|V_i\| \\
&\leq C(\frac{\sqrt{d}k^2 m}{\Delta\delta_v}\frac{w_{max}\text{cond}_2(V)^2}{w_{min}^2\sigma_{min}(V)^3})\|V_i\|\epsilon_z,
\end{aligned}
$$

for some universal constant $C$. Note that the last inequality used the assumption that $\epsilon_z$ is small enough. $\qquad\qquad\qquad\qquad\qquad\qquad\qquad\qquad\qquad\qquad\qquad\qquad\qquad\square$

*Proof.* (of Lemma 2.6) By definition, there exist some constants $\lambda$ and $\lambda'$ such that $\text{cond}_2(V_S) = \lambda'/\lambda$, and for all $w \in \mathcal{P}_{1,2}^k$, we have $\lambda \leq \|V_S w\| \leq \lambda'$. Note that each element of the factor $V_{S'}$ lies on the unit circle in the complex plane, then we have:

$$\lambda^2 \leq \|V_S w\|_2^2 \leq \|V_{S'} w\|_2^2 \leq (\lambda')^2 + \sqrt{k}d.$$

We can bound the condition number of $V_{S'}$ by:

$$\text{cond}_2(V_{S'}) \leq \sqrt{\frac{(\lambda')^2 + \sqrt{k}d}{\lambda^2}} = \sqrt{1 + \frac{\sqrt{k}d}{(\lambda')^2}}\text{cond}_2(V_S) \leq \sqrt{1 + \sqrt{k}}\text{cond}_2(V_S),$$

where the last inequality is because that $\max_w \|V_S w\|_2^2 \geq \|V_S e_1\|_2^2 = d$, we have $(\lambda')^2 \geq d$.

$\qquad\qquad\qquad\qquad\qquad\qquad\qquad\qquad\qquad\qquad\qquad\qquad\qquad\qquad\qquad\qquad\square$

*Proof.* (of Lemma 2.7) Denote $Y = \mathbb{E}_s[X_s]$. Note that $Y_{j,j} = 1$ for all diagonal entries. For $d = 1$ case, the point sources all lie on the interval $[-1, 1]$, we can bound the summation of the off diagonal entries in the matrix $Y$ by:

$$
\begin{aligned}
\sum_{j' \neq j} |Y_{j,j'}| &= \mathbb{E}_s[e^{i\pi<\mu^{(j')} - \mu^{(j)}, s>}] \\
&= \sum_{j' \neq j} e^{-\frac{1}{2}\pi^2\|\mu^{(j)} - \mu^{(j')}\|_2^2 R^2} \\
&\leq 2(e^{-\frac{1}{2}(\pi\Delta R)^2} + e^{-\frac{1}{2}(\pi(2\Delta)R)^2} + \cdots + e^{-\frac{1}{2}(\pi(k/2)\Delta R)^2}) \\
&\leq 2e^{-\frac{1}{2}(\pi\Delta R)^2}/(1 - e^{-\frac{1}{2}(\pi\Delta R)^2}) \\
&\leq \epsilon_x.
\end{aligned}
$$

For $d \geq 2$ case, we simply bound each off-diagonal entries by:

$$Y_{j,j'} = e^{-\frac{1}{2}\pi^2 \|\mu^{(j)} - \mu^{(j')}\|_2^2 R^2} \leq e^{-\frac{1}{2}\pi^2 \Delta^2 R^2} \leq \epsilon_x / k.$$

Apply Lemma 5.2 (Gershgorin's Disk Theorem) and we know that all the eigenvalues of $Y$ are bounded by $1 \pm \epsilon_x$. $\qquad\square$

*Proof.* (of Lemma 2.8) Let $\{X^{(1)}, \ldots, X^{(m)}\}$ denote the i.i.d. samples of the random matrix $X_s$ defined in (11), with $s$ evaluated at the i.i.d. random samples in $\mathcal{S}$. Note that we have

$$\|V_S w\|_2^2 = w^\top V_S^* V_S w = w^\top \left(\frac{1}{m} \sum_{i=1}^m X^{(i)}\right) w.$$

By definition of condition number, to show that $\mathrm{cond}_2(V_S) \leq \sqrt{\frac{1+2\epsilon_x}{1-2\epsilon_x}}$, it suffices to show that

$$(1 - 2\epsilon_x) I_{k \times k} \preceq \left(\frac{1}{m} \sum_{i=1}^m X^{(i)}\right) \preceq (1 + 2\epsilon_x) I_{k \times k}.$$

By Lemma 2.7, the spectrum of $\mathbb{E}_s[X_s]$ lies in $(1 - \epsilon_x, 1 + \epsilon_x)$. Here we only need to show that the spectrum of the sample mean $\left(\frac{1}{m} \sum_{i=1}^m X^{(i)}\right)$ is close to the spectrum of the expectation $\mathbb{E}_s[X_s]$. Since each element of the random matrix $X_s \in \mathbb{C}^{k \times k}$ lies on the unit circle in the complex plane, we have $X_s^2 \preceq k^2 I$ almost surely. Therefore we can apply Lemma 5.1 (Matrix Hoeffding) to show that for $m > \frac{k}{\epsilon_x} \sqrt{8 \log \frac{k}{\delta_s}}$, with probability at least $1 - \delta_s$, it holds that $\|\frac{1}{m} \sum_{i=1}^m X^{(i)} - \mathbb{E}_s[X_s]\|_2 \leq \epsilon_x$. $\qquad\square$

## 5  Discussions

### 5.1  Numerical results

We empirically demonstrate the performance of the proposed super-resolution algorithm in this section.

First, we look at a simple instance with dimension $d = 2$ and the minimal separation $\Delta = 0.05$. Our perturbation analysis of the stability result limits to small noise, i.e. $\epsilon_z$ is inverse polynomially small in the dimensions, and the number of measurements $m$ needs to be polynomially large in the dimensions. However, we believe these are only the artifact of the crude analysis, instead of being intrinsic to the approach. In the following numerical example, we examine a typical instance of 8 randomly generated 2-D point sources. The minimal separation $\Delta$ is set to be 0.01, and the weights are uniformly distributed in $[0.1, 1.1]$ The measurement noise level $\epsilon_z$ is set to be 0.1, and we take only 2178 noisy measurements ($\ll 1/\Delta^2$). Figure 1 shows reasonably good recovery result.

Figure 1: The xy plane shows the coordinates of the point sources: true point sources (cyan), the two closest points (blue), and the estimated points (red); the z axis shows the corresponding mixing weights. Dimension $d = 2$, number of point sources $k = 8$, minimal separation $\Delta = 0.05$ and the measurement noise level $\epsilon_z = 0.1$; we set the cutoff frequency $R = 200$ (in the same order as $1/\Delta$), take 2178 random measurements ($\ll 1/\Delta^2$).

Next, we examine the phase transition properties implied by the main theorem.

Figure 2 shows the dependency between the cutoff frequency and the minimal separation. For each fixed pair of the minimal separation and the cutoff frequency $(\Delta, R)$, we randomly generate $k = 8$ point sources in 4-dimensional space while maintaining the same minimal separation. The weights are uniformly distributed in $[0.1, 1.1]$. The recovery is considered successful if the error $\sum_{j \in [k]} \sqrt{\|\widehat{\mu}^{(j)} - \mu^{(j)}\|_2^2} \leq 0.1$ (on average it tolerates around $4\%$ error per coordinate per point source). This process is repeated 50 times and the rate of success was recorded. Figure 2 plots the success rate in gray-scale, where 0 is black and 1 is white.

We observe that there is a sharp phase transition characterized by a linear relation between the cutoff frequency and the inverse of minimal separation, which is implied by Theorem 2.2.

Figure 2: Fix dimension $d = 4$, number of point sources $k = 8$, number of measurements $m = k^2$, and the measurement noise level $\epsilon_z = 0.02$. We vary the minimal separation such that $\Delta$ ranges from 0.005 to 0.1, and we vary the cutoff frequency $R$ from 0 to 25. For each pair of $(\frac{1}{\Delta}, R)$ we randomly generate $k$ point sources and run the proposed algorithm to recover the point sources. The recovery is considered successful if the error $\sum_{j \in [k]} \sqrt{\|\widehat{\mu}^{(j)} - \mu^{(j)}\|_2^2} \leq 0.1$. This process is repeated 50 times and the rate of success was recorded.

In a similar setup, we examine the success rate while varying the minimal separation $\Delta$ and the number of measurement $m$.

In Figure 3, we observe that there is a threshold of $m$ below which the number of measurements is too small to achieve stable recovery; when $m$ is above the threshold, the success rate increases with the number of measurements as the algorithm becomes more stable. However, note that given the appropriately chosen cutoff frequency $R$, the number of measurements required does not depend on the minimal separation, and thus the computation complexity does not depend on the minimal separation neither.

Figure 3: Fix dimension $d = 4$, number of point sources $k = 8$, and the measurement noise level $\epsilon_z = 0.03$. We vary the minimal separation such that $\Delta$ ranges from 0.01 to 0.2, and we use the corresponding cutoff frequency $R = \frac{0.26}{\Delta}$. We also vary the number of measurements $m$ from 4 to 64. For each pair of $(\Delta, m)$ we randomly generate $k$ point sources and run the proposed algorithm to recover the point sources. The recovery is considered successful if the error $\sum_{j \in [k]} \sqrt{\|\widehat{\mu}^{(j)} - \mu^{(j)}\|_2^2} \leq 0.1$. This process is repeated 50 times and the rate of success was recorded.

## 5.2  Connection with learning GMMs

One reason we are interested in the scaling of the algorithm with respect to the dimension $d$ is that it naturally leads to an algorithm for learning Gaussian mixture models (GMMs).

We first state the problem: given a number of $N$ i.i.d. samples coming from a random one out of $k$ Gaussian distributions in $d$ dimensional space, the learning problem asks to estimate the means and the covariance matrices of these Gaussian components, as well as the mixing weights. We denote the parameters by $\{(w_j, \mu^{(j)}, \Sigma^{(j)})\}_{i \in [k]}$ where the mean vectors $\mu^{(j)} \in [-1, +1]^d$, the covariance matrices $\Sigma^{(j)} \in \mathbb{R}^{d \times d}$ and the mixing weights $w_j \in \mathbb{R}_+$. Learning mixture of Gaussians is a fundamental problem in statistics and machine learning, whose study dates back to Pearson[18] in the 1900s, and later arise in numerous areas of applications.

In this brief discussion, we only consider the case where the components are spherical Gaussians with common covariance matrices, namely $\Sigma^{(j)} = \sigma^2 I_{d \times d}$ for all $j$. Moreover, we define the separation $\Delta_G$ by:

$$\Delta_G = \frac{\min_{j \neq j'} \|\mu^{(j)} - \mu^{(j')}\|_2}{\sigma},$$

and we will focus on the well-separated case where $\Delta_G$ is sufficiently large. This class of well-separated GMMs is often used in data clustering.

By the law of large numbers, for large $d$, the probability mass of a $d$-dimensional Gaussian distribution tightly concentrates within a thin shell with a $\sqrt{d}\sigma$ distance from the mean vector. This concentration of distance leads to a line of works of provably learning GMMs in the well-separated

case, started by the seminal work of Dasgupta[6] (spherical and identical $\Sigma$, $\Delta_G \geq \Omega(d^{1/2})$, complexity $poly(d,k)$) and followed by works of Dasgupta & Schulman [8] (spherical and identical $\Sigma$, $d \gg \log(k)$, $\Delta_G \geq \Omega(d^{1/4})$, complexity $poly(d,k)$), Arora & Kannan [21] (general and identical $\Sigma$, $\Delta_G \geq \Omega(d^{1/4})$ complexity $O(k^d)$).

Instead of relying on the concentration of distance and use distance based clustering to learn the GMM, we observe that in the well-separated case the characteristic function of the GMM has nice properties, and one can exploit the concentration of the characteristic function to learn the parameters. Note that we do not impose any other assumption on the dimensions $k$ and $d$.

Next, we sketch the basic idea of applying the proposed super-resolution algorithm to learn well-separated GMMs, guaranteeing that $N$ the required number of samples from the GMM, as well as the computation complexity both are in the order of $poly(d,k)$. Since $\sigma$ is a bounded scalar parameter, we can simply apply grid-search to find the best match. In the following we assume that the $\sigma$ is given and focus on learning the mean vectors and the mixing weights.

Evaluate the characteristic function of a $d$ dimensional Gaussian mixture $X$, with identical and spherical covariance matrix $\Sigma = \sigma^2 I_{d \times d}$, at $s \in \mathbb{R}^d$:

$$\phi_X(s) = \mathbb{E}[e^{i<x,s>}] = \sum_{j \in [k]} w_j e^{-\frac{1}{2}\sigma^2 \|s\|_2^2 + i<\mu^{(j)},s>}.$$

Also we let $\widehat{\phi}_X(s)$ denote the empirical characteristic function evaluated at $s$ based on $N$ i.i.d. samples $\{x_1, \ldots x_N\}$ drawn from this GMM:

$$\widehat{\phi}_X(s) = \frac{1}{N} \sum_{l \in [N]} e^{i<x_l,s>}.$$

Note that $|e^{i<x_l,s>}| = 1$ for all samples, thus we can apply Bernstein concentration inequality to the characteristic function and argue that $|\widehat{\phi}_X(s) - \phi_X(s)| \leq O(\frac{1}{\sqrt{N}})$ for all $s$.

In order to apply the proposed super-resolution algorithm, define

$$f(s) = e^{\frac{1}{2}\sigma^2 \pi^2 \|s\|_2^2} \phi_X(\pi s) = \sum_{j \in [k]} w_j e^{i\pi<\mu^{(j)},s>}, \quad \text{and} \quad \widetilde{f}(s) = e^{\frac{1}{2}\pi^2\sigma^2 \|s\|_2^2} \widehat{\phi}_X(s).$$

In the context of learning GMM, taking measurements of $\widetilde{f}(s)$ corresponding to evaluating the empirical characteristic function at different $s$, for $\|s\|_\infty \leq R$, where $R$ is the cutoff frequency. Note that this implies $\|s\|_2^2 \leq dR^2$. Therefore, we have that with high probability the noise level $\epsilon_z$ can be bounded by

$$\epsilon_z = \max_{\|s\|_\infty \leq R} |f(s) - \widetilde{f}(s)| = O\left(\frac{e^{\sigma^2 dR^2}}{\sqrt{N}}\right).$$

In order to achieve stable recovery of the mean vector $\mu^{(j)}$'s using the proposed algorithm, on one hand, we need the cutoff frequency $R = \Omega(1/\sigma\Delta_G)$; on the other hand, we need the noise level $\epsilon_z = o(1)$. It suffices to require $\sigma^2 dR^2 = o(1)$, namely having large enough separation $\Delta_G \geq \Omega(d^{1/2})$. In summary, when the separation condition is satisfied, to achieve target accuracy in estimating the parameters, we need the noise level $\epsilon_z$ to be upper bounded by some inverse polynomial in the dimensions, and this is equivalent to requiring the number of samples from the GMM to be lower bounded by $poly(k,d)$.

Although this algorithm does not outperform the scaling result in Dasgupta[6], it still sheds light on a different approach of learning GMMs. We leave it as future work to apply super-resolution algorithms to learn more general cases of GMMs or even learning mixtures of log-concave densities.

## 5.3  Open problems

In a recent work, Chen & Chi [5] showed that via structured matrix completion, the sample complexity for stable recovery can be reduced to $O(k \log^4 d)$. However, the computation complexity is still in the order of $O(k^d)$ as the Hankel matrix is of dimension $O(k^d)$ and a semidefinite program

is used to complete the matrix. It remains an open problem to reduce the sample complexity of our algorithm from $O(k^2)$ to the information theoretical bound $O(k)$, while retaining the polynomial scaling of the computation complexity.

Recently, Schiebinger et al [22] studied the problem of learning a mixture of shifted and re-scaled point spread functions $f(s) = \sum_j w_j \varphi(s, \mu^{(j)})$. This model has the Gaussian mixture as a special case, with the point spread function being Gaussian point spread $\varphi(s, \mu^{(j)}) = e^{-(s-\mu^{(j)})^\top \Sigma_j^{-1}(s-\mu^{(j)})}$. We have discussed the connection between super-resolution and learning GMM. Another interesting open problem is to generalize the proposed algorithm to learn mixture of broader classes of nonlinear functions.

## 6 Auxiliary lemmas

**Lemma 6.1** (Matrix Hoeffding). *Consider a set $\{X^{(1)}, \ldots, X^{(m)}\}$ of independent, random, Hermitian matrices of dimension $k \times k$, with identical distribution $X$. Assume that $\mathbb{E}[X]$ is finite, and $X^2 \preceq \sigma^2 I$ for some positive constant $\sigma$ almost surely, then, for all $\epsilon \geq 0$,*

$$Pr\left(\left\|\frac{1}{m}\sum_{i=1}^{m} X^{(i)} - \mathbb{E}[X]\right\|_2 \geq \epsilon\right) \leq k e^{-\frac{m^2 \epsilon^2}{8\sigma^2}}.$$

**Lemma 6.2** (Gershgorin's Disk Theorem). *The eigenvalues of a matrix $Y \in \mathbb{C}^{k \times k}$ are all contained in the following union of disks in the complex plane: $\cup_{j=1}^{k} \mathcal{D}(Y_{j,j}, R_j)$, where disk $\mathcal{D}(a,b) = \{x \in \mathbb{C}^k : \|x - a\| \leq b\}$ and $R_j = \sum_{j' \neq j} |Y_{j,j'}|$.*

**Lemma 6.3** (Vector Random Projection). *Let $a \in \mathbb{R}^m$ be a random vector distributed uniformly over $\mathcal{P}_{1,2}^m$, and fix a vector $v \in \mathbb{C}^m$. For $\delta \in (0,1)$, we have:*

$$Pr\left(|<a, v>| \leq \frac{\|v\|_2}{\sqrt{em}}\delta\right) \leq \delta$$

*Proof.* This follows the argument of Lemma 2.2 from Dasgupta & Gupta [7]. Extension to complex number is straightforward as we can bound the real part and the imaginary part separately. □

## Footnotes

[3] This setting is different from the 2-norm separation condition. To see the difference, note that the toy algorithm does not work for constant shift $\mu^{(1)} = \mu^{(2)} + \Delta$. This issue is resolved in the general algorithm, when the condition is stated in terms of 2-norm separation.