[Reviews · NeurIPS 2015]

Submitted by Assigned_Reviewer_1

The paper analyzes the problem of super-resolving a collection of spikes from low-pass measurements. It introduces an algebraic approach based on tensor factorization. For k spikes, suggests taking about k log k random locations in Fourier space, plus d deterministic locations, and then sampling on a grid of size (k log k + d)^3 corresponding to all ways of summing three of these locations.

Because of the algebraic properties of Fourier measurements, the resulting 3-tensor is of rank k (up to noise). The paper proves that with high probability in the sample locations, the matrix of factors is well-conditioned, and hence the factors can be stably recovered using linear algebraic manipulations. These results are stable under small deterministic noise.

An attractive aspect of the approach is that the requirements do not depend exponentially on the signal dimension. There are two main downsides. The first is that the approach is suboptimal from the perspective of compressed sensing - it intrinsically requires k^3 measurements to recover k spikes. The second is that the stability result is limited to very small noise. This may be an artifact of the analysis, or may be intrinsic to this approach. It is somewhat difficult to judge, because numerical results are not provided. Comments in the response would be appreciated.

The main (clever) idea here is the reduction of the problem to 3-tensor decomposition; after this, the analysis follows in a natural way. The submission does not provide much basis for evaluating the practicality of the approach. However, from a theoretical perspective it obtains scalings (esp. in higher dimensions, where the number of samples required does not depend exponentially on d) that to my knowledge were not available using previous approaches.

Minor comments

In the comparison with the work of Candes and Fernandez-Granda [3], it should be mentioned that [3] allows general complex-valued amplitudes. The setup of this submission restricts the amplitudes to lie between zero and one. There are results of Candes and Morgenshtern on the arxiv for this setup, which might be a more relevant point of comparison.

The analysis of noisy tensor decomposition forces the noise level to be very small -of \ell^\infty norm about 1/k^4. An advantage is that the noise can be arbitrary. Can the authors comment: (a) is this restriction an artifact of the tensor decomposition algorithm adopted here and its analysis, or an intrinsic limitation to the proposed approach, and (b) can a better stability result be obtained with stochastic (say, Gaussian) noise?

The submission appears to have been written in a hurry. Please spellcheck. Also, "One useful fact..." [line 060] is false. I understand what is intended, but that kind of a typo that early on doesn't build confidence.

It most applications that this reviewer is aware of, d is a small constant, and so measuring \Delta in L^2 vs L^\infty does not make a substantial difference.
Summary: The paper introduces an algebraic approach to super resolving point sources from low-pass measurements. It proves that about k^3 measurements suffice, independent of the signal dimension d. The algorithm is based on 3-tensor decomposition, with a cleverly chosen set of measurements.

Author Feedback
Author rebuttal: We thank all the reviewers for the positive comments.

In the following, we shall mainly address the detailed comments from the Assigned Reviewer 1.

Major comments:
" There are two main downsides.
The first is that the approach is suboptimal from the perspective of compressed sensing - it intrinsically requires k^3 measurements to recover k spikes.

The second is that the stability result is limited to very small noise. This may be an artifact of the analysis, or may be intrinsic to this approach. It is somewhat difficult to judge, because numerical results are not provided. "

Response:
In the revised version, we have improved the dependence of required number of measurements to O((k+d)^2) instead of O((k+d)^3). We agree with the reviewer that this approach is still not yet information theoretic optimal in terms of required number of measurements. We are working towards further reducing this scaling to be linear in k and d, while maintaining stable recovery and low computational complexity.

The stability result is a consequence of our analysis of the simple method for factorizing the low rank 3rd tensor, and our main message is to show that it has polynomial scaling and thus "stable recovery".

In practice, after setting up the well-conditioned (guaranteed) 3rd order low rank tensor with the noisy measurements, more sophisticated tensor decomposition methods should be applied to retrieve the model parameters, which are known to be more noise tolerant, especially under stochastic noise model.

However, getting a sharper bound for tensor decomposition is not the main focus of this work. The stability of the proposed approach will be demonstrated by numerical experiments in the longer version.

Minor comments:
"The setup of this submission restricts the amplitudes to lie between zero and one. There are results of Candes and Morgenshtern on the arxiv for this setup, which might be a more relevant point of comparison."

Response:
We thank the reviewer for pointing this out and suggesting the reference. Actually we can extend our setup to allow complex-valued amplitudes, and this follows from the uniqueness of complex-valued tensor decomposition. We will revise the draft accordingly to make the setup more general.

Minor comments:
"The submission appears to have been written in a hurry. Please spellcheck. Also, "One useful fact..." [line 060] is false."

Response:
Thank you for pointing this mistake out. We have fixed it and will carefully proofread the draft.

Minor comments:
"It most applications that this reviewer is aware of, d is a small constant, and so measuring \Delta in L^2 vs L^\infty does not make a substantial difference."

Response:
We agree with the reviewer that when d is a small constant, Delta measured in different norms does not differ by much.

In the longer version, we will apply this approach to learning mixture of Gaussians, where d is the dimension of the distribution and can be very high depending on the application. In this application, it is important that we use the separation Delta in L_2 instead of L_infty.

Moreover, even in the regime where d is small, our results still outperforms the existing methods in terms of required number of measurements and the runtime.